# Play to Learn: A Game to Improve Seismic-Risk Perception

Maria Grazia Filomena [1], Bruno Pace [2], Massimo De Acetis [1], Antonio Aquino [1], Massimo Crescimbene [3], Marina Pace [4] and Francesca Romana Alparone [1,*]

1 Department of Neuroscience, Imaging and Clinical Science, 66100 Chieti, Italy
2 Department of Engineering and Geology, INGEO, 60013 Chieti, Italy
3 Istituto Nazionale di Geofisica e Vulcanologia, INGV, 90146 Palermo, Italy
4 Department of Psychology, The New School, New York, NY 10011, USA
* Correspondence: francesca.alparone@unich.it; Tel.: +39-871-355-6593

**Abstract:** A board game designed by psychologists and geologists to improve seismic-risk perception is presented. In a within-subjects repeated-measure study, 64 Italian high-school students rated their perception of seismic risk in relation to the hazard, vulnerability and exposure of the area in which they lived, before and after the game. A repeated-measures analysis of variance (ANOVA), which considered perception of seismic risk as the dependent variable and time as the independent variable, revealed that the board game affected the dependent variable, particularly the perception of hazard and vulnerability. The results confirm the effectiveness of the game in changing participants' seismic risk perception, properly because the game was built with consideration of the variables that make up seismic risk. The board game proved to be an effective and fun educational tool to be used in future earthquake risk prevention programs.

**Keywords:** seismic risk perception; educational board game; earthquakes

## 1. Introduction

The number of places across the world highly exposed to the possibility of natural disasters, such as earthquakes, is high. Earthquakes are responsible for about one third of global economic and human natural-disaster-related losses [1]. In the last 30 years, approximately 1000 earthquakes were the primary cause of worldwide human losses (about 800,000 known fatalities) and the second leading cause of worldwide economic losses (USD 844 billion) due to natural disasters. Yet, people's ability to adequately perceive seismic risk remains insufficient. Interventions most often focus on engineering measures and geo-physical interventions and, more rarely, on human determinants of behaviors aimed at prevention. In this regard, Slovic [2] highlights that people evaluate hazards via an intuitive judgment called risk perception, which is a subjective prediction of the probability of damage based on personal beliefs, judgments, and opinions. The implementation of risk prevention behaviors further requires proper perception of all dimensions of earthquake risk from a multidisciplinary perspective. The present work contributes to this goal.

Seismic risk is generally defined as a combined function of three parameters: (a) hazard, the probability of a seismic event occurring over a period of time and of a given intensity; (b) vulnerability, the degree of structural resistance against damage due to ground shaking; and (c) exposure, a value reflecting the number of people and buildings in the assessed area, together with the number of commercial activities, industries and infrastructures [3]. In brief, seismic risk deals with the potential comprehensive effect of an earthquake on the territory and depends on the respective underlying natural characteristics, the resilience of buildings and structures, and the presence of people and activities.

However, risk seems to take on different meanings for people. Research on risk perception is typically divided into two fields: the psychometric paradigm [4] and the cultural-theory paradigm [5]. The psychometric paradigm, which is the dominant approach,

asks people to judge the risks and benefits associated with certain activities or events in order to identify the mental strategies or heuristics that people use to make risk judgments [4]. The cultural-theory paradigm, on the other hand, focuses on how risks are collectively and culturally represented and developed. Research that has taken a psychometric approach has shown that there is a gap between expert and population judgments [6,7]. In the Italian context, for example, which is a medium–high seismic-risk area, scientists of the National Institute of Geophysics and Volcanology (INGV) investigated the level of risk perception of the Italian population [8] using an ad-hoc seismic-risk-perception scale that assesses the three dimensions of seismic risk (hazard, vulnerability, and exposure). The study involved 5585 citizens from all the Italian regions who were asked to evaluate the seismic risk of their area of residence. The evaluations of participants were then compared to the seismic-risk rate assigned by experts to Italian' territories, called "hazard by law" (Law No. 3274 in the Official Gazette No. 105 of 8 May 2003). Hazard by law specifies the seismic-hazard level of Italy's municipalities as one of four areas: area 1 includes municipalities with the highest probability of being affected by strong earthquakes; area 2 includes municipalities with a medium–high probability of being affected by an earthquake of a low intensity; area 3 includes municipalities with a medium probability of being affected by moderate earthquakes; and area 4 has a low probability of being affected by an earthquake. The results of the study [8] showed that nine out of ten citizens living in high-seismic-risk areas underestimated the potential danger of earthquakes with respect to that indicated by "hazard by law". Regarding this underestimation, Slovic et al. [9] stated that when people judge a particular risk, they rely on what they have seen, remembered or heard about the risky event rather than on statistical data [9], so that we talk about risk perception with an emphasis on its subjective aspect.

Why is risk perception so important? The literature on this topic has mainly shown its positive correlation with disaster preparedness, i.e., with the implementation of emergency planning activities or the promotion of protective behavior [10–12]. Secondly, risk perception indicates the way in which people estimate the probability of a certain event occurring and the extent of expected damage [13]. This approximation guides people's behavior towards adopting or not adopting risk mitigation strategies, such that people with high-risk perception are more likely to implement risk-reduction behavior than those with low-risk perception [14–17]. Some authors [10,18,19] have also stressed that risk perception is a complex process involving both cognitive and affective aspects. Among the affective processes, the role of negative emotions has often been emphasized, especially fear, which are induced, above all, by the severity of consequences [20]. It has been pointed out, however, that affective processes only partly overlap with cognitive risk assessment [21,22]. In support of this is the fact that a positive relationship between risk perception, preparedness and fear is not always present. Indeed, some studies, for example, Qing et al. [23], have found a positive relationship between these variables, while others have found no relationship at all [24–26].

To sum up, the extent to which people perceive risk assumes an important role in motivating them to avoid, mitigate, or adapt to hazards. As mentioned, because risk perception is often inaccurate, it is important to align the subjective perception of risk with the objective risk indicated by experts. In what way can this goal be achieved in the field of seismic risk? The Sendai Framework for Disaster Risk Reduction [27], the international document adopted by UN member states during the World Conference on Disaster Risk Reduction, reports that any action to address natural disasters should have schools as its main target, as schools may play a central role in the prevention of non-structural hazards (such as furnishing and equipment, electrical and mechanical fixtures, and architectural features). Educating people from a young age about risks means increasing the likelihood that they will develop damage-reduction behaviors that will be carried over into adulthood [28]. However, when comparing school curricula in compulsory schools in four European countries (Italy, Spain, Portugal and Iceland), Bernhardsdottir et al. [29] found that none of the schools examined offered a specific course

on earthquake and volcano education. In a similar study, Komac, Zorn and Ciglič [30] revealed that school textbooks (in 36 European countries) pay little attention to earthquakes (and floods) compared to volcanoes, to which much more attention is devoted. These studies indicate that schools underestimate their role in natural-hazard education, and that there is a need for an increase in educational programs related to earthquakes, as well as for thinking of new tools to actively engage students. Further, starting from a common background, educational programs on seismic-risk perception focus on the unique characteristics of seismic risk (i.e., hazard, vulnerability and exposure) rather than on the common characteristics of natural hazards.

The literature provides some examples of how educational programs on seismic risk should engage students. For example, in Turkey, Mermer et al. [31] implemented and evaluated an interactive earthquake education program in a middle school, which included 52 informative training sessions accompanied by interactive components (such as placing stickers on seismic maps) and animated videos. The study revealed that knowledge of earthquakes and disaster preparedness actions increased significantly after the intervention. Another example is provided by an Italian educational project, called "EDURISK" [32], aimed at children of ages 4 to 13. EDURISK was promoted and supported by the Department of Civil Defense to foster a "risk culture" of dealing with earthquakes and other natural phenomena through prevention. The methodology provides teachers with tools such as informational booklets and books with fun vignettes to create classroom pathways on earthquake and volcano knowledge. EDURISK volunteers—a multidisciplinary staff of research, communication, and education professionals—produced and distributed these materials for students and organized training for teachers. The EDURISK project involved many students (70,000) and teachers (4000) and has been active since 2002.

The value of these initiatives has been their use of engaging methods and tools to disseminate understanding of the earthquake phenomenon, prompting each student to interact with the teaching materials. As Schulz et al. noted [33], it is important to motivate students because if the motivation to learn is lacking, the information gained is unlikely to impact behavior. Thus, it is important that we focus on activities that sustain learning. It would be appropriate to think about teaching methods that can arouse curiosity and interest, to engage students in hands-on activities that enable them to stimulate reasoning and integration of information. The literature [34–37] indicates that play can achieve this: the use of games in a lesson, as part of teaching, motivates students to participate and creates a positive attitude toward learning [38,39]. The importance of play as a tool for cognitive and social development was also emphasized by Piaget [40], who considered play a fundamental tool in helping children move from one stage of development to the next.

Starting from this premise, we designed and realized an original board game, called Ruaumoko, intended to be offered to schools to support teaching about seismic hazards, providing: (a) knowledge of the characteristics of a physical and built environment which define its degree of seismic risk based on the factors of hazard, vulnerability, and exposure; (b) flexibility of thinking, i.e., the ability to reason about new information, link it to previous beliefs and develop new beliefs and attitudes; and (c) awareness of proper damage prevention and mitigative actions. We chose a game because, by nature, it engages and stimulates curiosity and interest, and is a fun way to communicate knowledge on the topic. However, our goal was to provide information about the earthquake phenomenon but to do this in an engaging way. Moreover, given the importance of risk perception in guiding people's protective behaviors, a second goal was to intervene on subjective risk perception. To achieve these goals, in constructing our board game, we were guided by the parameters that constitute seismic risk (hazard, vulnerability and exposure). The present work aims to test the effectiveness of the designed board game in achieving our goals. This project is a collaboration between psychologists and geologists to provide a new method to support the teaching of natural hazards, especially earthquakes, and to improve the perception of earthquake risk. To test the effectiveness of Ruaumoko in increasing players' perception

of earthquake risk and supporting learning, we conducted a study with a within-subjects repeated-measure design involving students from an Italian scientific high school.

## 2. Materials and Methods

### 2.1. Study Instrument: Ruaumoko

The name Ruaumoko is taken from Māori mythology: Ruaumoko is the god of earthquakes and volcanoes. The design, layout, and logic of Ruaumoko are inspired by the well-known board game Monopoly. The Ruamoko box consists of a game board, a die, four boxes of cards, five pieces (representing players), and 75 houses of different colors (representing buildings). The game board is structured as follows: in the center is a map of a fictitious city with geophysical and seismotectonic characteristics similar to those of the central Apennines in Italy, around the map are square spaces on which to move throughout the game, below the squares is a legend outlining the characteristics of the territories, and at the bottom of the board is a space on which to place the game cards (Figure 1). The game board, logic and colored houses are designed to demonstrate to players that (a) the degree of hazard is determined above all by the geophysical and seismotectonic characteristics of the territory, and that this variable is fundamental to proper urban design and building construction; (b) the higher the quality of buildings, the lower the likelihood of them being damaged; and (c) crowding buildings and people can lead to more damage as a result of an earthquake. The game cards are divided into four sets, labelled "situation," "chance," "earthquakes" and "knowledge". Game cards are designed to cause players to lose buildings (punishment) when (a) players construct buildings in highly hazardous areas, (b) they construct buildings with high vulnerability, or (c) they construct buildings in areas defined as overcrowded. Players do not lose buildings (reinforcement) when they correctly consider the characteristics of the territory while constructing their buildings or redevelop their buildings (trade multiple high-vulnerability buildings for a single low-vulnerability building).

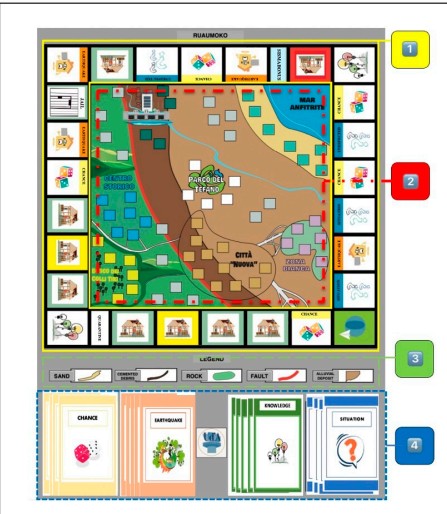

**Figure 1.** Ruaumoko game (1. Quadrants board, 2. Map, 3. Legend, 4. Cards).

### 2.2. How to Play Ruaumoko

Ruaumoko is played by a minimum of 3 to a maximum of 5 players and takes place over two rounds. The player who has collected the most buildings at the end of two rounds wins. Buildings of different colors indicate different degrees of vulnerability: green for high vulnerability, yellow for medium vulnerability, and red for low vulnerability. At the beginning of the game, each player can construct one green, one yellow, and one red building on the map. Next, pieces are placed on the "go" box and each player, in turn, rolls the die. Based on the score on the die, the pieces are moved along the path; the course of

the game is completely random. When a piece happens upon a space marked "situation," "chance," or "earthquakes," the player whose turn it is must take a card from the relevant deck, read it aloud and follow the directions. When the piece happens upon a space marked "knowledge," however, players do not read the card aloud but keep what they read to themselves. When the piece lands on a space reserved for building construction, players place a house on the map respecting the color indicated on the space (see Appendix A). The aim of the game is to maintain the buildings one constructs over time: this goal can be achieved if they invest in building quality (red buildings) over quantity (green buildings, which are more common and easier to build).

*2.3. Participant and Procedure*

To obtain an estimate of the number of students needed, an a-priori power analysis was conducted using G*Power 3.1 [41] adopting a repeated-measures ANOVA within-subject method, which revealed that at least 54 participants were required to achieve a medium effect size (f = 0.25). A total of 64 students (32 boys, 31 girls, 1 agender, age 13.85 years, SD = 0.46) attending their first year at a scientific high school in Pescara (Abruzzo, central Italy, with medium–high level of seismic risk, source INGV) were enrolled for the study with parental consent. Before running the game and collecting data, we made sure that there had not yet been specific teaching on natural disasters and earthquakes within the curricula of the classes involved. Participation in the research was completely voluntary and it was made clear in advance that participation in the research would not affect students' grades in any way. We gave the players who won the game a reward. All students completed all phases of assessment. Before starting the game, the researchers explained the rules of the game to the students and clarified any doubts. They also remained available in the classroom for the duration of the game. Research was conducted across two school days, with a one-week gap between the first and second sessions. During session 1 (T1—baseline) participants were asked to complete a questionnaire that included questions about socio-demographic information (age, gender, class and nationality) and a first measure of risk perception (details of the measures in the following paragraph). The questionnaire was completed in class and took about 20 min. After completing the questionnaire, participants were randomly assigned to 14 teams (4 or 5 players each) and played Ruaumoko for about an hour. The game was played during class time in the presence of the teacher. One week later (session 2—post-game), participants filled out the same version of the questionnaire to record the change in risk perception At the end, a debriefing was conducted to collect impressions and evaluations about the game: players had the opportunity to discuss, with the experimenters (psychologists and geologists), topics related to the dynamics of the game, ask for clarification about earthquakes and seismic risk, and discuss among themselves. Figure 2.

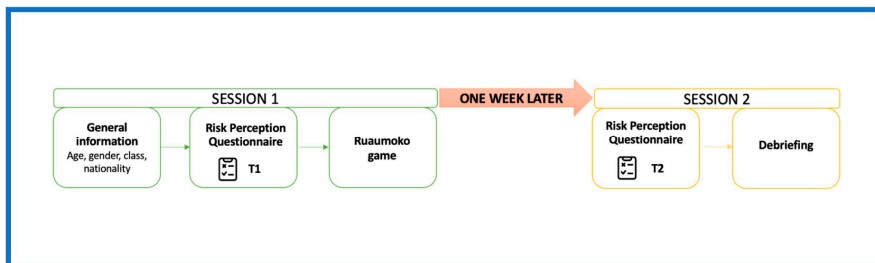

**Figure 2.** Timeline of the experiment.

*2.4. Measures*

To measure risk perception, we used Risk Perception Questionnaire (RPQ) [42], a semantic differential [43] consisting of bipolar adjectives (e.g., near—far) separated by a seven-point scale, referring to the three components (subscales) of earthquakes risk: hazard, vulnerability, and exposure. The hazard subscale measures one's perception of earthquake

risk for the area in which they live. It consists of 10 bipolar adjectives linked to the question "Imagine an earthquake in the area where you live, how do you describe it?" A single score for hazard was calculated as a mean of the 10 adjectives ($\alpha = 0.74$) with the highest value representing the greatest degree of hazard perception. The vulnerability subscale measures perception of resilience in the event of an earthquake with respect to one's home and the school they attend. It consists of 12 bipolar adjectives linked to the question "Compared to an earthquake, how do you envision your home/school?". A single score for hazard was calculated as mean of the 12 adjectives ($\alpha = 0.84$) with the highest value representing the greatest degree of hazard perception. The exposure subscale ($\alpha = 0.59$) measures the extent of perceived potential harm attributed to the event of an earthquake in one's area. It consists of 7 bipolar adjectives linked to the question "Compared to an earthquake, how do you describe the area where you live?". A single score for hazard was calculated as mean of the 7 adjectives ($\alpha = 0.54$) with the highest value representing the greatest degree of exposure perception. A total RPQ score was obtained by the sum of the score of each RPO subscale: hazard, vulnerability, and exposure.

### 3. Results

To test the increase in risk perception following participation in the game, an analysis of variance for repeated measures (ANOVA) was performed considering earthquake risk perception as the dependent variable and time as the independent variable (T1, T2). The results showed a significant effect of time on the RPQ, $F_{(1.63)} = 9.03$; $p < 0.001$; and $\eta p^2 = 0.11$, confirming our hypothesis that playing the Ruaumoko game increases players' perception of earthquake-related risk (Figure 3a). To understand whether the game acted upon all three subscales of the RPQ, we conducted three separate ANOVAs with time as an independent variable and hazard, vulnerability and exposure as dependent variables, respectively. ANOVAs showed a significant effect of time for the hazard subscale, $F_{(1.63)} = 7,54$; $p < 0.001$; $\eta p^2 = 0.11$, and vulnerability, $F_{(1.63)} = 5.96$; $p = 0.017$; $\eta p^2 = 0.09$, but no significant effect for the exposure subscale, $F_{(1.63)} = 2.35$; $p = 0.13$; $\eta p^2 = 0.00$. (Figure 3b).

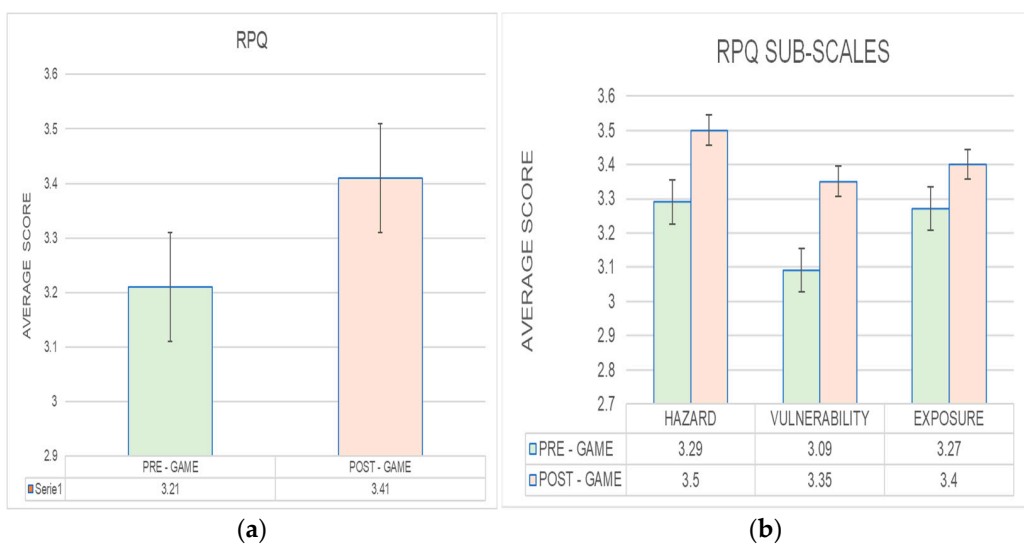

**Figure 3.** (**a**) ANOVAs results and average scores between pre- and post-game; (**b**) ANOVAs results and average scores between pre- and post-game for hazard, vulnerability and exposure factors.

### 4. Discussion

*4.1. Improving the Perception of Earthquake Risk*

The purpose of the present study was to demonstrate the effectiveness of a board game—designed by psychologists and geologists—in improving secondary-school students' perception of earthquake risk. Our results confirmed the hypothesis by showing that a

single game session can produce a significant change in players' RPQ scores. Ruaumoko, thus, proved to be a useful tool to increase the perception of earthquake risks in an active and engaging way.

This result is in-line with the literature on "learning by doing," according to which active student involvement increases motivation to learn [44,45] when compared to traditional teaching, which pushes students into a passive role. Our findings show that Ruamoko has a significant effect on hazard and vulnerability, but not on exposure. In this regard, we can say that the game produced a change in the perception of hazard precisely because the dynamics of the game prompted players to make their choices after having adequately reflected on the different effects an earthquake can have on buildings in relation to the different geological compositions of the sites on which they are constructed. Similarly, for vulnerability, the players had to think carefully about the different effects an earthquake can have if building designs are not adapted to the ground characteristics or are not constructed appropriately. We believe this was achieved because the game-based learning method has the capacity to recreate situations that are dangerous or difficult to actually reproduce, allowing players to get to the heart of the subject matter in question [46]. Regarding exposure, no significant change was found. It is worth considering the possibility that players may have had difficulty grasping this dimension. Indeed, quantifying the value of a loss of buildings and people in each area is a difficult task, even outside of the game. Furthermore, while hazard and vulnerability are directly proportional, exposure does not necessarily have to be. In future studies, we hope to improve Ruaumoko by better targeting the exposure factor. Overall, the results suggest that Ruaumoko is effective in promoting better perception of seismic risk and that the game is a suitable tool for the topic of seismic risk. It is worth emphasizing that a proper perception of seismic risk makes it possible to impose earthquake-proof criteria and implement policies for forecasting and securing buildings, exponentially reducing possible damages, and making citizens more aware of the characteristics of the area in which they live. Young people, when equipped with good risk perception, have the potential to develop good risk-response skills, not just for their own safety but for that of the members of their community as well [47].

### 4.2. The (Board) Game Is a Valuable Tool for Educational Projects on Seismic-Risk Perception

The role of educational institutions and local governments in promoting and funding educational projects aimed at this end is of paramount importance [48]. To our knowledge, this is the first educational project that aimed to act directly on the perception of seismic risk, where previous projects have focused only on behavioral outcomes regarding the right behaviors to adopt during, before or after an earthquake [49–52]. For this reason, we believe this work is important for the purpose of providing educational institutions with a simple and inexpensive tool to educate the population, from a young age, on the correct perception of earthquake risk, forestalling fears related to the perceived unpredictability and uncontrollability of earthquake damage. Our results allow us to reassess the potential offered by board games, even during a period in which computer games are becoming increasingly popular. Board games can offer students an opportunity to study in a different way, allowing them to be actively involved in study activities [53]. Moreover, in contrast to electronic devices, board games foster the development of social skills through interaction among players [54]. We firmly believe that the use of board games deserves attention in education, because they allow us to work on ethical (e.g., rule-following), interpersonal (e.g., group interactions), and cognitive (e.g., abstract thinking) skills, as well as to increase curiosity and interest. In our opinion, future research activities in the field of environmental risks should consider the potential of board games for the added reason that it is a tool which does not require a large financial investment (unlike games that use virtual reality and/or apps for PCs, tablets, or smartphones) and can be applied easily and without the necessity for the active guidance of teachers. Although our results are very satisfactory overall, it is necessary to replicate the effectiveness of Ruaumoko in areas with different degrees of seismic risk, as well as in different school grades.

## 5. Conclusions

We designed and implemented a new board game, called Ruaumoko, concerning seismic risk for 11- to 16-year-olds, with the aim of providing an easy-to-apply tool in school and non-school settings to effectively intervene on the perception of seismic risk and, in turn, foster and spread a proper culture of damage prevention. The effectiveness of the game was tested on a high-school population in an area of central Italy with an objective medium level of seismic risk. Comparing the RPQ scores before and after the game experience, a significant increase in the perception of seismic risk was found, given by the increase in RPQ scores, as a result of correctly learning the factors that determine the impact of an earthquake on a built environment. Our research provides evidence that innovative and engaging tools can be used to promote the prevention and management of natural hazards.

**Author Contributions:** Conceptualization, M.G.F. and F.R.A.; methodology, M.G.F., F.R.A. and B.P.; validation, B.P., F.R.A. and M.G.F.; formal analysis, M.G.F. and A.A.; investigation, M.G.F., M.D.A., A.A. and B.P.; resources, M.G.F. and F.R.A.; data curation, A.A.; writing—original draft preparation, M.G.F.; writing—review and editing, F.R.A., B.P., A.A., M.G.F., M.P. and M.C.; visualization, F.R.A., M.G.F. and B.P.; supervision, F.R.A. and B.P. All authors have read and agreed to the published version of the manuscript.

**Funding:** This research received no external funding.

**Institutional Review Board Statement:** The study was conducted in accordance with the Declaration of Helsinki.

**Informed Consent Statement:** Informed consent was obtained from all subjects involved in the study.

**Data Availability Statement:** https://www.dropbox.com/sh/5kiuej8b8jw63jn/AACFbTkdIgUJfIteUH1tX6lwa?dl=0, accessed on 20 February 2023.

**Acknowledgments:** We sincerely thank the scientific high school Galileo-Galilei of Pescara, in particular Massimiliano Nerone and Francesca Santeusanio. We were allowed to enter the school despite the COVID-19 restrictions. We thank students attending 1°A, 1°I and 1°F for they active participation and for the positive feedback about our game. Moreover, we thank Asia Palma for having accompanied, helped and supported us during the different phases of the design and implementation of the game.

**Conflicts of Interest:** The authors declare no conflict of interest.

## Appendix A

The map was designed to best reflect the meaning of seismic hazard: different soil types (sand, rock, alluvial deposits, cemented debris) and faults are reported on the map and players must choose whether, where and how to locate housing, public and commercial buildings (place the houses in the squares drawn on the map). Considering the vulnerability indicator, we introduced buildings of different colors to indicate different degree of vulnerability: green indicates high vulnerability, yellow medium vulnerability and red low vulnerability. When an earthquake occurs during the game (using the "earthquake" cards), players tend to lose more green and yellow buildings than red ones. For exposure parameters, we did not add any relevant graphics on the map but incorporated this indicator into the dynamics of the game itself. For example, we included unforeseen events during the game, where players who chose to receive constructed buildings in highly crowded areas were penalized with the loss of constructed buildings.

The board has 28 squares: 9 reserved for buildings, where the player can freely decide where to build on the map; 15 squares reserved for cards, where the player can draw from the deck of cards indicated by the square; 1 square depicting quarantine, where the player will be stationary for the current turn; 1 square depicting prison, where the player will be stationary for the current and next turn; 1 square reserved for a seismic bonus, where players can give up a high vulnerability building in favor of a low vulnerability one; and r1 starting square. "Situation cards" are unexpected events that occur during the game

with the purpose of making players who have not considered risk indicators lose buildings when constructing buildings (e.g., "If you've built yellow or green buildings near a fault you have the obligation to retrofit one of those buildings. You lose two green ones to retrofit it as yellow and one yellow one to retrofit it as red. If you don't have enough buildings, you lose one"). "Chance cards" allow players to construct new buildings or receive them as gifts. However, gifted buildings can be constructed in dangerous or overcrowded areas of the map (e.g., "You have the chance to construct a building in the fault's proximity without losing anything"). In addition, these cards also offer the possibility of exchanging one's own buildings for another player's. "Earthquake cards" cause earthquakes of varied intensities to occur and are intended to cause players to lose buildings based on the area in which they were built along with the type of building they opted for (e.g., "There has been an earthquake which has caused soil liquefaction, meaning that the soil/sand becomes "quicksand". If you've constructed green or yellow buildings on top of sandy soil, you lose them"). Earthquake cards include the "Big-One," which is the most violent earthquake that can occur during the game and affects all players. "Knowledge cards" contain information about seismic risk and the properties of the game map. The purpose of these cards is to provide knowledge about seismic risk in general and, at a more specific level, to provide information that can be used to build well during play (e.g., "The shaking of buildings is not the same everywhere but rather depends on the conditions of the local territory. In general, it will be greatest wherever the terrain is soft (sands and alluvial plains), and least on rigid terrain (rock, detritus)").

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
