# Peer review of "Play to Learn: A Game to Improve Seismic-Risk Perception"

_sustainability, doi:10.3390/su15054639_

Round 1

Reviewer 1 Report

In the present manuscript, the authors reported on the effects of a board game on improving risk perception of earthquake. The tool was well designed, and the study and the data were well presented. The study has important applied implications for hazard prevention and control. Several issues can be improved before publishing.

1. While the study focuses on seismic risk perception, it would be helpful to move the definition of risk perception (the third paragraph on p.2) to an earlier section.

2. How is seismic risk perception training different from earthquake or other more general risk training program?

3. In 2.3 Participant and procedure (p.5), the total number of participants is 64. However, there were 32 male and 31 female participants. Was there any other gender category? Please double check the number, or clarify the inconsistency.

4. Please divide the Discussion into a few several shorter paragraphs. 

Reviewer 2 Report

The manuscript entitled "Play to learn: a game to improve seismic risk perception" by Filomena et al. submitted to sustainability represents an interesting contribution to seismic risk communication introducing an experience with the use of a board game to educate on seismic risk. Very interesting it is also the evaluation of the experience's impact on the pupil's risk perception.

I have found the manuscript well-organized and well-written. The figures are clear and well explicative. References are well-considered and integrated into the manuscript. The presented data, discussions, and conclusion of the paper are well-grounded. For this reason, I approve of its publication. 

Reviewer 3 Report

The manuscript is important in terms of developing a practical application on how seismic risk perception can be developed in adolescents. However, in the article, the perception of risk and other concepts related to this subject (such as fear, preparedness) need to be clarified further. In this regard, the current literature needs to be reviewed further. For example; It is suggested that the authors benefit from the study titled "Examination of Risk Perception, Fear and Preparedness of Individuals Experiencing Earthquakes" on the subject.
